# Progressive Context-Aware Aggregation Network Combining Multi-Scale and Multi-Level Dense Reconstruction for Building Change Detection

**Chuan Xu** [1,†] , **Zhaoyi Ye** [1,†] , **Liye Mei** [1,†] , **Wei Yang** [2] , **Yingying Hou** [1] , **Sen Shen** [3] , **Wei Ouyang** [2] **and Zhiwei Ye** [1,*]

1   School of Computer Science, Hubei University of Technology, Wuhan 430068, China
2   School of Information Science and Engineering, Wuchang Shouyi University, Wuhan 430064, China
3   School of Weapon Engineering, Naval Engineering University, Wuhan 430032, China
*   Correspondence: hgcsyzw@hbut.edu.cn
†   These authors contributed equally to this work.

**Abstract:** Building change detection (BCD) using high-resolution remote sensing images aims to identify change areas during different time periods, which is a significant research focus in urbanization. Deep learning methods are capable of yielding impressive BCD results by correctly extracting change features. However, due to the heterogeneous appearance and large individual differences of buildings, mainstream methods cannot further extract and reconstruct hierarchical and rich feature information. To overcome this problem, we propose a progressive context-aware aggregation network combining multi-scale and multi-level dense reconstruction to identify detailed texture-rich building change information. We design the progressive context-aware aggregation module with a Siamese structure to capture both local and global features. Specifically, we first use deep convolution to obtain superficial local change information of buildings, and then utilize self-attention to further extract global features with high-level semantics based on the local features progressively, which ensures capability of the context awareness of our feature representations. Furthermore, our multi-scale and multi-level dense reconstruction module groups extracted feature information according to pre- and post-temporal sequences. By using multi-level dense reconstruction, the following groups are able to directly learn feature information from the previous groups, enhancing the network's robustness to pseudo changes. The proposed method outperforms eight state-of-the-art methods on four common BCD datasets, including LEVIR-CD, SYSU-CD, WHU-CD, and S2Looking-CD, both in terms of visual comparison and objective evaluation metrics.

**Keywords:** building change detection; remote sensing; deep learning; progressive context-aware aggregation; dense reconstruction

## 1. Introduction

Building change detection (BCD) is one of the most significant research directions in remote sensing image processing [1]. By identifying structures at the same geographical location in multiple images at various time periods, it is possible to determine whether substantial changes have occurred to the buildings in the area, where substantial change refers to changes in physical attributes, such as the conversion of wasteland into buildings and converting roads into buildings [2,3]. BCD is widely used in land resource management [4], environmental monitoring [5], urban planning [6], and post-disaster reconstruction [7]. Therefore, it is of the utmost importance to develop effective BCD methods.

Generally, there are two types of BCD methods: traditional methods and deep learning (DL)-based methods. Traditional methods are further divided into pixel- and object-based methods [8]. Pixel-based methods usually generate difference maps by comparing spectral

or texture information between pixels, and then obtain BCD results by threshold segmentation or clustering algorithms [9,10]. However, independent pixel information ignores contextual information, which leads to a lot of noise [11]. Moreover, pixel-based methods are mainly suitable for low-resolution images with simple detail information [12]. Hay et al. [13] introduced the concept of objects to remote sensing images. There has been a significant amount of research conducted on object-based methods since then [14–16]. Based on the rich spectral, texture, structural, and geometric information in bi-temporal images, the core idea is to segment images into unrelated objects and analyze the differences [17]. By utilizing objects' spectral and spatial characteristics, object-based methods can improve detection accuracy [18]. The effectiveness of these methods, however, depends on the object segmentation algorithm, and does not take semantic information into account. This can easily be disrupted by pseudo-variation [19]. Therefore, the generalization performance of these two types of methods is not very suitable to meet realistic needs due to limitations of applicable image pixels and specific conditions [20,21].

With the continuous improvement of satellite earth observation capabilities, it becomes easier to obtain remote sensing images with high spatial and temporal resolution [22–24]. Meanwhile, details of ground objects are revealed more effectively [25]. In recent years, DL has demonstrated excellent results in a variety of computer vision tasks [26–30]. Compared to traditional methods, DL technology not only improves feature extraction ability, but also improves detection efficiency, so it is widely used in BCD as well [31,32]. Because convolutional neural networks (CNN) process large amounts of data, methods such as SNUNet [33], STANet [34], and SCADNet [35] have performed reasonably well on BCD. By cleverly fusing the shallow and deep features of convolutional neural networks in a suitable manner, it is possible to improve the BCD performance of high-resolution remote sensing images [36]. Despite CNN's ability to provide deep features rich in semantic information, which are conducive to identifying internal differences in buildings, it lacks more detailed information [37]. In addition, while deep convolution can extract the details of shallow features, thereby preserving the building edge contour, an inadequate amount of semantic information will result in error detection [38]. Compared to CNN, Transformer is capable of extracting feature information and modeling global dependency structures, which can reduce the probability of feature information being lost during model calculation [39]. Chen et al. [40] transform the input bi-temporal images into high-level semantic markup and model the context in a compact tag-based space-time model. Utilizing the relationship between each pixel and the semantic information enhances the feature representation of the original pixel space, further highlighting the changing buildings. To achieve Transformer's ideal state of generalizing existing models to solve other problems, a large number of high-quality datasets must be trained on the network [41,42].

In spite of the fact that many researchers have proposed excellent BCD methods in recent years, the following two problems still persist. On the one hand, current mainstream methods produce relatively simple feature forms that do not adequately represent the characteristics of changing buildings, so they are easily influenced by pseudo-changes. On the other hand, there are large scale differences in the extracted feature information. Ignoring the fusion of heterogeneous features will result in the loss of valuable information, which makes it difficult to accurately identify local–global feature expressions between buildings. Therefore, we propose the progressive context-aware aggregation network for BCD. The critical goal of our proposed method is to extract the local–global changing building features effectively, and fuse the extracted multi-scale and multi-level features more reasonably. Finally, we use a fully convolutional network (FCN) to further refine the feature map after dense fusion to obtain BCD results. Our major contributions are summarized below:

(1) We design the progressive context-aware aggregation module to cleverly stack deep convolution and self-attention, thus leveraging the feature extraction capability of the above two individuals. Deep convolution extracts shallow change information about buildings in bi-temporal images, while self-attention further acquires high-

level semantic information. As a result, our extracted local–global feature information contains not only local useful information but also global complementary information.

(2) We propose the multi-scale and multi-level dense reconstruction (MMDR) module, which groups extracted local–global features according to pre- and post-temporal sequences and gradually reconstructs them, making the local–global information fusion more reasonable. Each group is connected through our multi-level dense reconstruction strategy. In addition, subsequent groups are able to reconstruct information based on prior reconstruction information provided by the previous group. This promotes the retention of effective information during the reconstruction process, and further enhances the ability to recognize areas that are changing within the building.

## 2. Materials and Methods

In Section 2, we first describe the proposed method briefly, followed by the progressive context-aware aggregation and multi-scale and multi-level dense reconstruction modules in detail, and finally introduce the hybrid loss function.

### 2.1. Network Architecture

Figure 1 illustrates the structure of our method, which receives bi-temporal images as input. Our first step is to increase bi-temporal image channels using CBGBlock. CBGBlock is a compound block that includes 3 × 3 convolution, BatchNorm layer, and GELU activation. Then, we use the progressive context-aware aggregation module to extract local–global feature information from pre- and post-temporal images, respectively. The MMDR module enables us to connect local–global feature information appropriately. Finally, the FCN prediction head outputs the BCD results.

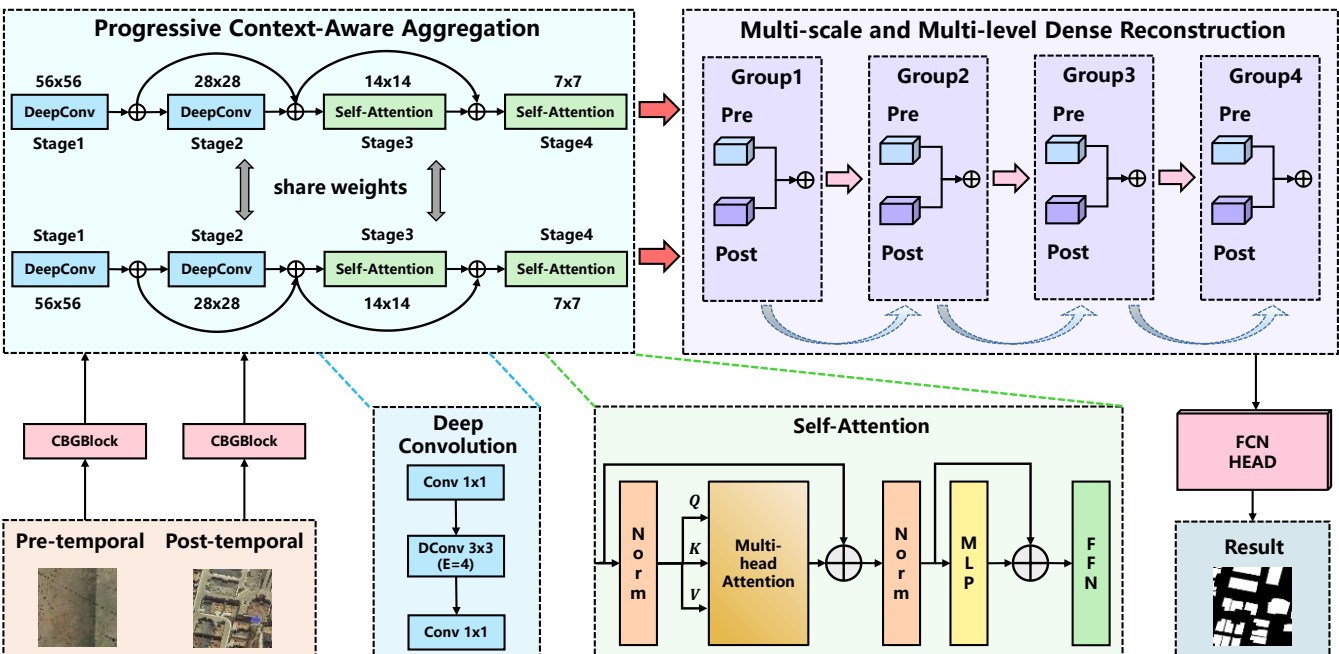

**Figure 1.** Overview of the proposed method. Firstly, bi-temporal images pass through the CBGBlock, which increases the number of channels while ensuring image resolution. After that, our progressive context-aware aggregation module utilizes deep convolution to extract local features, while self-attention is used to derive global features based on local features. The MMDR module can group the extracted sequential feature information and intensively connect it to the corresponding feature information. Finally, our FCN prediction head outputs the BCD result.

### 2.2. Progressive Context-Aware Aggregation

In the bi-temporal images, the buildings vary in size and shape, and there is a large amount of noise in the form of clouds, lights, and shadows on buildings. For a better description of the variation in buildings, we utilize the progressive context-aware aggregation module with a Siamese structure to obtain local–global feature information. Our Siamese structure consists of a two-channel structure with shared weights, which enables deep convolution to capture local features related to changing buildings, primarily containing low-level semantic information related to small buildings; based on the local feature information, the self-attention mechanism is able to extract global feature information rich in deep semantic information.

To collect local feature information among a small range of receptive fields, deep convolution can be performed using a convolution kernel of fixed size:

$$out_i = \sum_{j=1}^{N} W_{i-j} \odot in_i \tag{1}$$

where $in_i$ and $out_i$ represent the input and output at pixel $i$, respectively. $W_{i-j}$ denotes the kernel of deep convolution. $N$ represents all pixels in the local neighborhood at pixel $i$. In our implementation, $N = 9$, which corresponds to the convolution domain of a $3 \times 3$ convolution.

Based on the local information, to further enhance the context-awareness of the extracted features, we use self-attention to extract long-range information. Self-attention is capable of collecting global feature information in a wide range of perceptual fields, and its weight is based on $in_i$ and $in_j$ in a dynamic manner:

$$out_i = \sum_{j \in \mathbb{S}} \frac{\exp(in_i^T \cdot in_j)}{\sum\limits_{k \in \mathbb{S}} \exp(in_i^T \cdot in_k)} in_j \tag{2}$$

where $\mathbb{S}$ represents the global spatial space, $\sum\limits_{k \in \mathbb{S}} \exp(in_i^T \cdot in_k)$ denotes the dynamic attention weight.

The final result can be calculated by combining Equations (1) and (2) as follows:

$$out_i^{pre} = \sum_{j \in \mathbb{S}} \frac{\exp(in_i^T \cdot in_j + W_{i-j})}{\sum\limits_{k \in \mathbb{S}} \exp(in_i^T \cdot in_k + W_{i-k})} in_j \tag{3}$$

$$out_i^{post} = \sum_{j \in \mathbb{S}} \left( \frac{\exp(in_i^T \cdot in_j)}{\sum\limits_{k \in \mathbb{S}} \exp(in_i^T \cdot in_k)} + W_{i-j} \right) in_j \tag{4}$$

where $out_i^{pre}$ and $out_i^{post}$ represent the addition of the local static deep convolution kernel to the global dynamic attention matrix before and after softmax normalization, respectively. As simple as these Equations appear, their output depends on both the static weight $W_{i-j}$ and the dynamic inputs ($in_i^T \cdot in_j$), ensuring that the network learns more about the changing information progressively.

It is pertinent to note that one of the main differences between deep convolution and self-attention is the size of the receptive field. An expanded receptive field is generally associated with more contextual information. However, the BCD task entails not only large building changes, but also a significant number of smaller changes that cannot be captured by the large receptive field [43]. At the same time, deep convolutions are more effective at extracting low- and mid-level semantic information in the initial stages of multi-scale feature extraction. Therefore, as shown in Table 1, we construct our progressive context-aware aggregation module by stacking deep convolution and self-attention.

**Table 1.** Specific parameter description of the progressive context-aware aggregation module.

| Stages | Kernel Size | Attention Head | Blocks | Channels | Spatial Size |
|---|---|---|---|---|---|
| Stage1-DeepConv | 3 | - | 2 | 96 | 1/2 |
| Stage2-DeepConv | 3 | - | 3 | 192 | 1/4 |
| Stage3-Self-Attention | - | 32 | 5 | 384 | 1/8 |
| Stage4-Self-Attention | - | 32 | 2 | 768 | 1/16 |

### 2.3. Multi-Scale and Multi-Level Dense Reconstruction

Due to the significant scale difference between the extracted local and global feature information, fusing the feature information directly will result in an incorrect alignment and connectivity between the various features, which in turn will lead to a loss of key information during the fusion process, increasing the probability of false detection. In order to maximize the value of these feature information, we propose the MMDR module as shown in Figure 2, which groups the local–global feature information according to pre- and post-temporal groups. Specifically, to fully take advantage of the features from all groups, multi-level dense reconstructions are introduced between every group to allow for all subsequent groups to fuse the features learned by any prior groups.

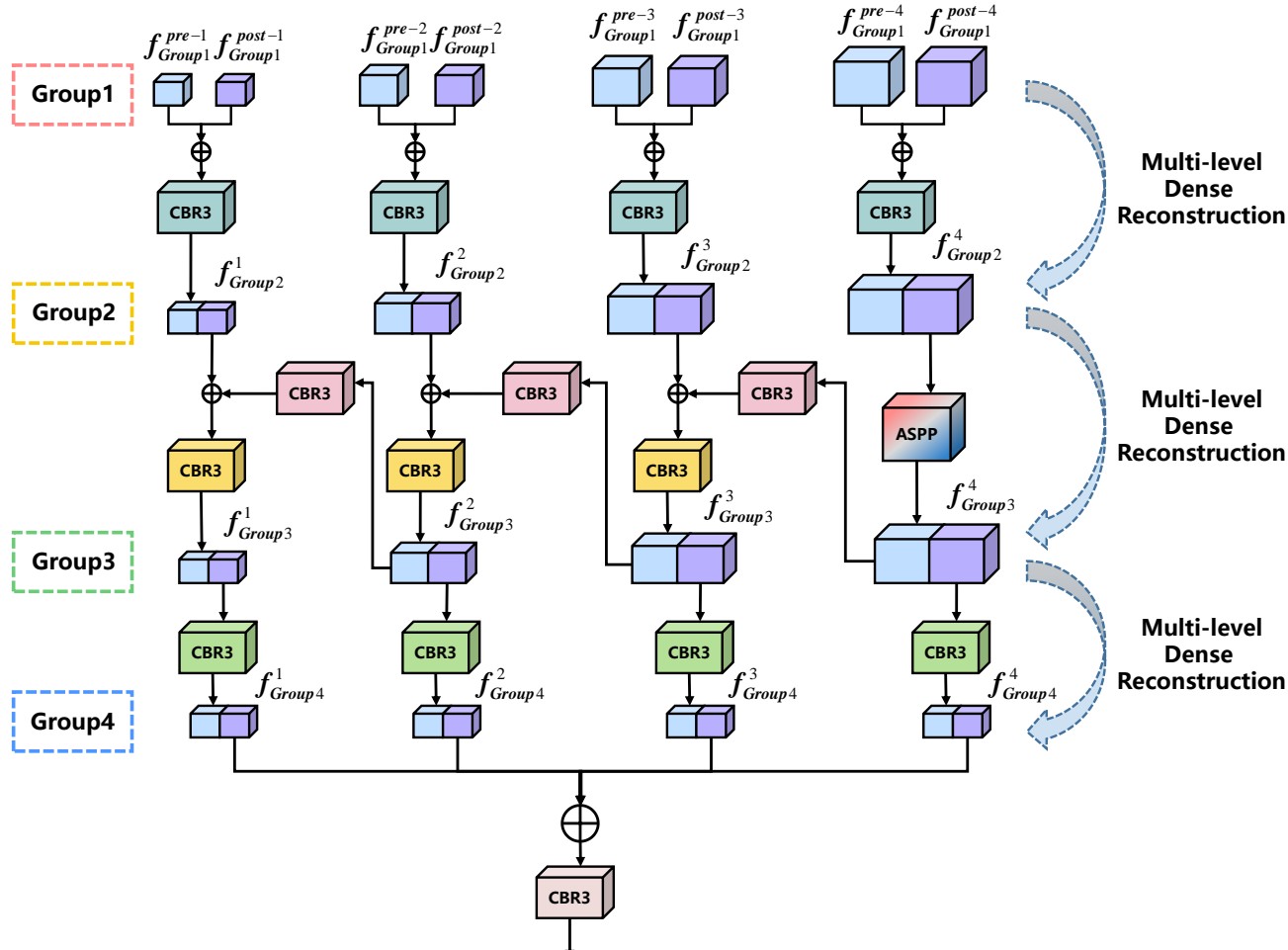

**Figure 2.** Overview of the multi-path and multi-scale dense reconstruction module.

As a result of the Siamese structure of the progressive context-aware aggregation module, eight feature maps of different sizes can be generated from bi-temporal images, four of which are pre-temporal feature maps and the remaining four are post-temporal

feature maps. Based on the following formula, we can obtain the second group of local–global feature maps:

$$f^i_{Group2} = CBR_3(f^{pre-i}_{Group1} \oplus f^{post-i}_{Group1}) \tag{5}$$

where $\oplus$ denotes the Concatenation operation, which cascades multi-scale extracted local–global features on the channel dimensions. $CBR_3$ is a composite block that contains a $3 \times 3$ convolution kernel with a stride of 1, a BatchNorm layer, and a ReLU activation. In the second group, the output channels of the four local–global feature maps are 96, 192, 384, and 768, respectively.

By injecting cavities into convolution blocks, Atrous Spatial Pyramid Pooling (ASPP) [44] is capable of expanding the convolution receptive field, capturing more details of building changes. As a result, we use ASPP for $f^4_{Group2}$ with the most channels in order to obtain $f^4_{Group3}$. Specifically, the third group of local–global feature maps can be obtained by the following calculations:

$$f^i_{Group3} = \begin{cases} ASPP(f^i_{Group2}), i = 4 \\ CBR_3(f^i_{Group2} \oplus CBR_3(f^{i+1}_{Group3})), 1 \le i \le 3 \end{cases} \tag{6}$$

where ASPP consists of a $CBR_3$ block with a convolution kernel size of $1 \times 1$, as well as three $CBR_3$ blocks with injection cavities. Each $CBR_3$ block has a dilation of 6, 12, and 18. Following Adaptive Pooling, a $CBR_3$ block with a convolution kernel size of $1 \times 1$ produces an output. A next step is to combine the output from the above processes with the $\oplus$ operation and then to use a $CBR_3$ block to calculate $f^i_{Group3}, i = 1, 2, 3$. In the third group, the output channels of the four local–global feature maps are 96, 192, 384, and 768, respectively.

For further refinement of the feature map information, we perform the $CBR_3$ again on the third group of feature maps to reduce the output channels to 96, thus obtaining the fourth group of local–global feature maps:

$$f^i_{Group4} = CBR_3(f^i_{Group3}) \tag{7}$$

The final dense fusion feature map is obtained by performing the $\oplus$ operation on all the feature maps in the fourth group and then performing the $CBR_3$ operation:

$$OUT = CBR_3(f^1_{Group4} \oplus f^2_{Group4} \oplus f^3_{Group4} \oplus f^4_{Group4}) \tag{8}$$

### 2.4. Loss Function

Due to the fact that most of the pixels in the bi-temporal images correspond to unchanged building areas, we adapt the BCE loss [5] to address the problem of imbalanced samples for BCD. BCE loss is defined as follows:

$$L_{BCE} = CD_{G_i} \log(CD_{P_i}) + (1 - CD_{G_i}) \log(1 - CD_{P_i}) \tag{9}$$

where $CD_{P_i}$ and $CD_{G_i}$ represent the values of pixel $i$ in the predicted image and the GT image, respectively.

It is evident that a small value of $CD_{P_i}$ in a changed class is accompanied by a large value of $CD_{G_i}$. In order to alleviate this issue, we use an additional DICE loss [5], which is calculated as follows:

$$L_{DICE} = 1 - \frac{2 * (CD_{P_i} \cap CD_{G_i})}{CD_{p_i} + CD_{G_i}} \tag{10}$$

where $CD_{P_i}$ and $CD_{G_i}$ represent the values of pixel $i$ in the predicted image and the GT image, respectively. Therefore, our hybrid loss is shown as follows:

$$L = L_{BCE} + L_{DICE} \tag{11}$$

## 3. Experiments and Results

We first introduce the four BCD datasets (LEVIR-CD, SYSU-CD, WHU-CD, S2Looking-CD) used in the experiments in detail, followed by the evaluation indicators and parameter settings. After that, we analyze the ablation experiment results. As a final step, we compare our method with eight other comparison methods for comprehensive visual and quantitative experiments.

### 3.1. Datasets

To demonstrate the effectiveness of our proposed method, we use four common BCD datasets: LEVIR-CD, SYSU-CD, WHU-CD, and S2Looking-CD. Each dataset contains bi-temporal images as well as building labels that are changing over time. The specifics of their introduction are as follows:

1.  The LEVIR-CD [34] is a collection of architectural images created by Bei-hang University, containing original Google Earth images collected between 2002 and 2018. Every original image has a resolution of 0.5 m and is 1024 × 1024 pixels in size. These changes involve barren land, residential areas, garages, grasslands, and other building modifications. In order to facilitate faster computation, each image was divided into 256 × 256 pixels without overlap. We, therefore, used 3096 images for training, 432 pairs for validation, and 921 pairs for testing. Figure 3 illustrates six different scenarios from the LEVIR-CD dataset.

2.  The SYSU-CD [39] dataset was released by Sun Yat-Sen University. A total of 20,000 256 × 256 pixels with a resolution of 0.5 m aerial images were captured between 2007 and 2014 in Hong Kong. The construction of urban and mountain buildings, urban roads, and coastline expansion comprise the majority of the changes in the dataset. The images were divided into training, validation, and testing sets according to a ratio of 6:2:2, resulting in 12,000, 4000, and 4000 images for training, validation, and testing, respectively. Figure 4 illustrates six different scenarios from the SYSU-CD dataset.

3.  The WHU-CD [41] dataset is a BCD dataset released by Wuhan University and contains one image of 15,354 × 32,507 pixels with a resolution of 0.5 m. The original image pairs were taken between 2012 and 2016 in Christchurch. The reconstruction of buildings after earthquakes and the transformation of wasteland into buildings are the two major types of building change. We cropped the original image to 256 × 256 pixels without overlapping, and obtained 7432 images in total. Based on the 7:1:2 ratio, we divided the cut images into three sets, the training set, the validation set, and the test set, with 5201, 744 and 1487 images, respectively. Figure 5 illustrates six different scenarios from the WHU-CD dataset.

4.  The S2Look-CD [45] dataset consists of 5000 bi-temporal very high-resolution images taken from three types of satellites, Gaofen Satellite (GF), SuperView Satellite (SV), and Beijing-2 Satellite (BJ-2), between 2017 and 2020, with a wider perspective to provide richer information about changes. The imaging area covers a wide range of rural areas throughout the world with a variety of complex features. There are 1024 × 1024 pixels in each image, and the image resolution is 0.5~0.8 m. We cropped each image into 512 × 512 pixels with a 50% overlap on each side (256 pixels for horizontal and vertical, respectively) to obtain 45,000 images. Our next step was to divide the images according to the ratio of 7:1:2, resulting in 31,500, 4500, and 9000 pairs of images for training, validation, and testing, respectively. Figure 6 illustrates six different scenarios from the S2Looking-CD dataset.

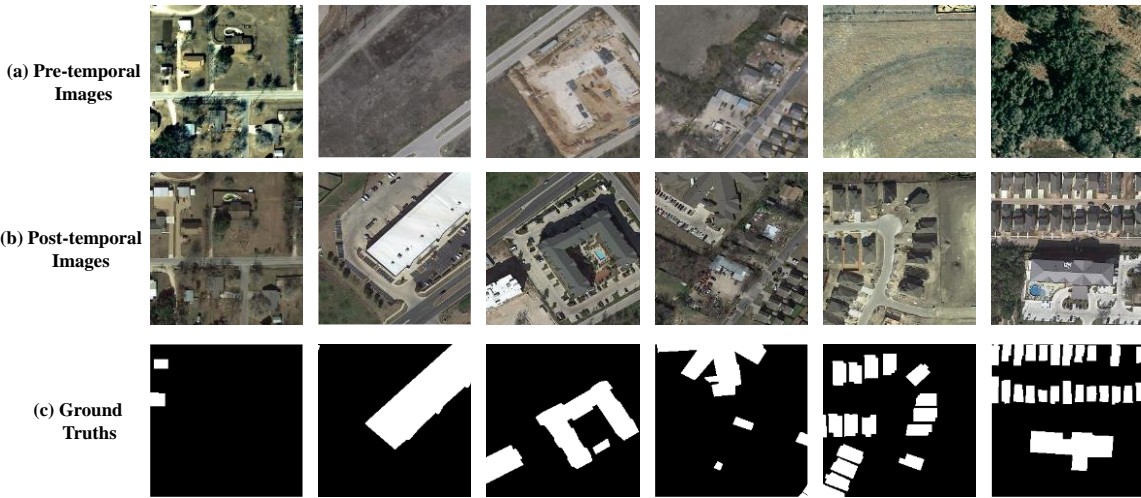

**Figure 3.** A demonstration of six scenarios from the LEVIR-CD dataset.

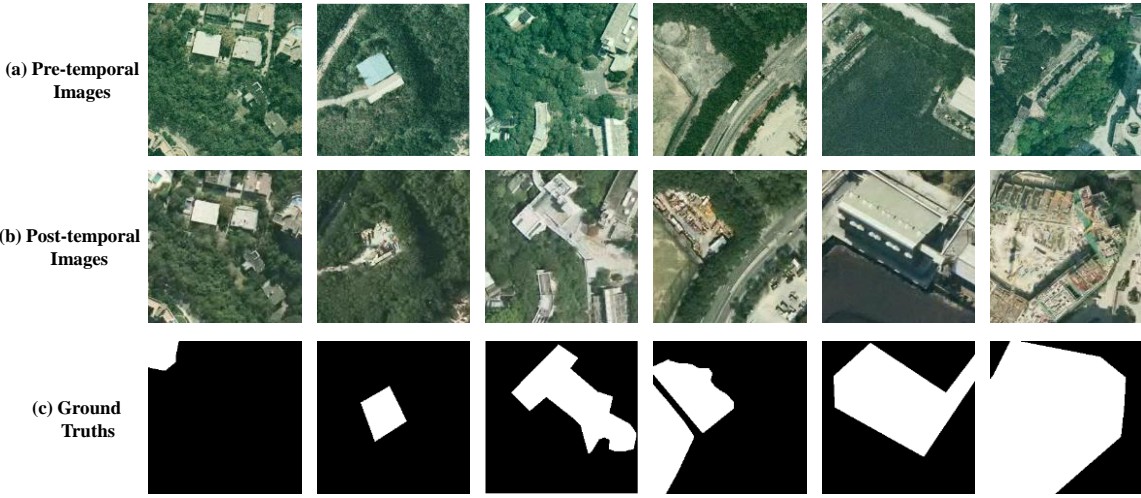

**Figure 4.** A demonstration of six scenarios from the SYSU-CD dataset.

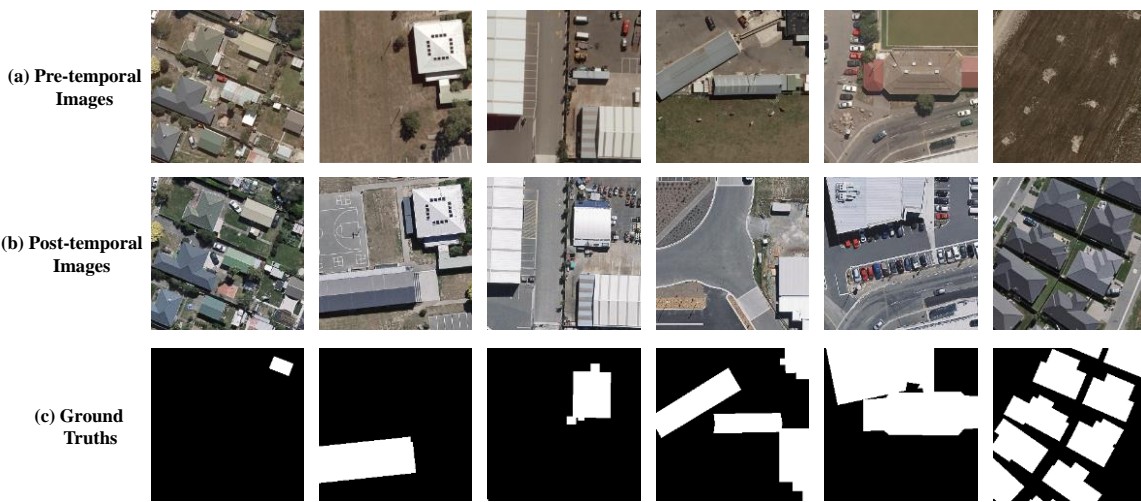

**Figure 5.** A demonstration of six scenarios from the WHU-CD dataset.

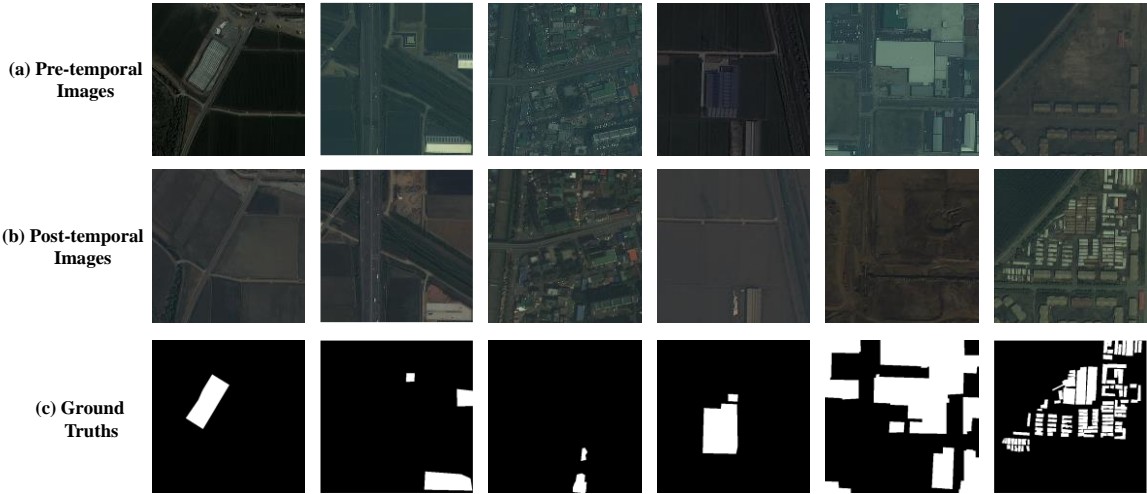

**Figure 6.** A demonstration of six scenarios from the S2Looking-CD dataset.

*3.2. Experimental Details*

3.2.1. Evaluation Metrics

In order to evaluate the CD methods, we use the following eight metrics: Precision, Recall, F1-score, mIOU, OA, and Kappa. In these metrics, a higher Precision denotes a lower false detection rate, whereas a higher Recall indicates a lower miss detection rate. F1-score, mIOU, OA, and Kappa values range from 0 to 1, with higher values representing a stronger performance. Furthermore, we consider IOU_0 and IOU_1, which represent IOU for constant pixels and changing pixels, respectively. To be more specific, we calculate evaluation metrics as follows:

$$\text{Precision} = \frac{TP}{TP + FP} \tag{12}$$

$$\text{Recall} = \frac{TP}{TP + FN} \tag{13}$$

$$\text{F1-score} = \frac{2 \times TP}{2 \times TP + FP + FN} \tag{14}$$

$$\text{IOU\_0} = \frac{TN}{TN + FP + FN} \tag{15}$$

$$\text{IOU\_1} = \frac{TP}{TP + FP + FN} \tag{16}$$

$$\text{mIOU} = \frac{\text{IOU\_0} + \text{IOU\_1}}{2} \tag{17}$$

$$\text{OA} = \frac{TP + TN}{TP + TN + FP + FN} \tag{18}$$

$$\text{PE} = \frac{(TP + FP)(TP + FN) + (FP + TN)(FN + TN)}{(TP + TN + FP + FN)^2} \tag{19}$$

$$\text{Kappa} = \frac{\text{OA} - \text{PE}}{1 - \text{PE}} \tag{20}$$

According to the formula above, $TP$ stands for True Positive, $FP$ stands for False Positive, $TN$ stands for True Negative, and $FN$ stands for False Negative. Note that $PE$ is an intermediate variable in the calculation of Kappa.

3.2.2. Parameter Settings

All of our experiments are conducted using the Pytorch DP framework, which provides high performance computing. A single NVIDIA Tesla A100 GPU is used with a GPU memory of 80 GB. During model training, the batch size is set to 24, and the maximum training epoch for each model is 400. We utilize AdamW as an optimizer with an initial learning rate of 0.00035 to avoid a small learning weight, and a weight decay rate of 0.001. In order to prevent overfitting during training, we use the early stop method.

*3.3. Ablation Experiment*

Table 2 shows the results of ablation experiments, which evaluate the effectiveness of the progressive context-aware aggregation and MMDR modules. Due to ASPP's ability to improve the receptive field, the F1-score of the network can be increased from 89.71% to 90.84% when ASPP is added to the baseline. As a result of adding the MMDR module to the baseline, our performance has been further enhanced, and its F1-score has reached 91.65%. This is because in a dense reconstruction strategy, prior feature information can be taken into consideration as part of the fusion process, resulting in more abundance of feature information.

**Table 2.** The results of ablation experiments on the LEVIR-CD dataset.

| Method | Precision | Recall | F1-Score | IOU_0 | IOU_1 | mIOU | OA | Kappa |
|---|---|---|---|---|---|---|---|---|
| Baseline | 91.93 | 87.59 | 89.71 | 97.54 | 81.34 | 89.44 | 97.78 | 88.47 |
| Baseline + ASPP | 91.40 | 90.29 | 90.84 | 97.77 | 83.22 | 90.50 | 97.99 | 89.71 |
| Baseline + MMDR | 92.49 | 90.83 | 91.65 | 97.97 | 84.59 | 91.28 | 98.17 | 90.62 |
| Ours (C-C-C-C) | 92.52 | 90.75 | 91.62 | 97.96 | 84.54 | 91.25 | 98.17 | 90.59 |
| Ours (C-C-C-T) | 92.41 | **91.19** | 91.80 | 98.00 | 84.83 | 91.42 | 98.20 | 90.78 |
| Ours (C-T-T-T) | 92.56 | 90.81 | 91.68 | 97.98 | 84.63 | 91.31 | 98.18 | 90.66 |
| **Ours (C-C-T-T)** | **93.41** | 90.67 | **92.02** | **98.07** | **85.22** | **91.64** | **98.26** | **91.04** |

Note that the values in bold are the highest.

In order to better understand the effectiveness of deep convolution and self-attention embedding in our progressive context-aware aggregation module, we examine the effects of four different embedding modes (C-C-C-C, C-C-C-T, C-T-T-T, C-C-T-T), where C represents deep convolution and T represents self-attention. The results of the experiments indicate that simple deep convolution (C-C-C-C) has the least ideal performance among the four, but its Precision value can still reach 92.52%. By replacing the last layer of deep convolution with self-attention (C-C-C-T), the Recall value is significantly improved, reaching 91.19%. Due to the self-attention mechanism, global change information can be captured effectively, reducing missed detections. Based on this, we try to substitute all the last three layers with the self-attention mechanism (C-T-T-T), but the results are not satisfactory, with F1-score and Recall indexes of 91.68% and 90.81%, respectively. This is because only one layer of deep convolution cannot fully extract shallow local variation features. Therefore, the advantages of the self-attention mechanism are not maximized to obtain robust global changing features. Finally, when we replace the second layer with deep convolution (C-C-T-T), our F1-score and mIOU are both the highest, at 92.02% and 91.64%, respectively. As a result, we have established that reasonable embedding deep convolution and self-attention can significantly enhance the performance of BCD.

*3.4. Visual Comparative Experiments*

We select eight popular CD methods for visual and quantitative comparison experiments in order to demonstrate the effectiveness of the proposed method, including four fully convolutional-based methods (FC-EF [46], FC-Siam-conc [46], FC-Siam-diff [46], CDNet [47]), one LSTM-based method (LUNet [48]), two Transformer-based methods (IFNet [49], and BITNet [40]), and one CNN-Transformer combined method (MSCANet [50]).

Figure 7 illustrates the BCD results of various methods for five different scenarios using the LEVIR-CD dataset, including small building changes, medium building changes, large building changes, and dense building changes. Our method detects all real changes without false positives in almost all scenarios. Despite only a small building change in the scene of the first row, the change in texture is not significant, resulting in poor results for other methods, while our method detects the main body of the changing building, as well as its boundary information. Due to illumination effects, all three FC-based methods, CDNet, LUNet, and MSCANet, miss one building change in the fourth row. Some edges are hidden in shadows, making it difficult to determine the exact boundary. Although all methods detect the actual change region well in the scene in the fifth row, it is clear that the building edges extracted by the other eight methods have a significant number of false detected regions, whereas our method extracts more accurate details of the building edges.

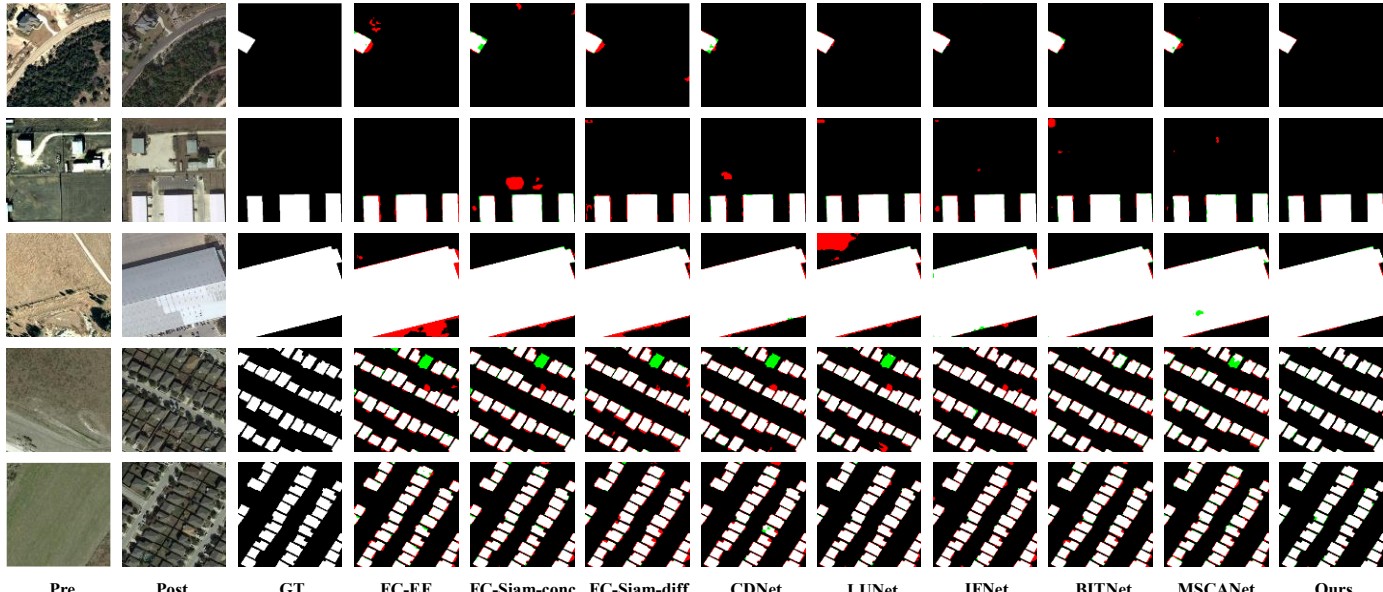

**Figure 7.** Visualization results of different CD methods in five scenarios from the LEVIR-CD dataset. We plot the FP in red and the FN in green in order to better compare the CD results.

Figure 8 illustrates the BCD results for various methods on the SYSU-CD dataset. In the first row of Figure 1, although our method has some missed detection areas, there is only a small amount of false detection. In contrast, the false detections of FC-Siam-diff, IFNet, BITNet, and IFNet are particularly prominent, misjudging road and vegetation changes as building changes, respectively. Due to the fact that buildings are being constructed, most methods perform poorly in the second and third rows. Despite the presence of false and missing detections in the results of our method, they are less than in other methods. Because the actual change labels have a sinuous texture, all methods cannot effectively extract the building edge information in the fourth row. Additionally, CDNet, LUNet, IFNet, BITNet, and MSCANet all incorrectly identify vegetation changes as building changes on the left side. In the fifth row, FC-Siam-conc incorrectly detects the change in vegetation as a change in buildings, resulting in a large area of red false positive pixels, while our method has the lowest number of false detections.

Figure 9 displays the visualized BCD results for the five scenes we collected from the WHU-CD dataset. There has been a drastic change in the bi-temporal images in the first row. As a result, CDNet, LUNet, IFNet, and BITNet have been unable to produce optimal detection results. LUNet incorrectly detects parking spaces as building changes. BITNet does not produce false detections, but it completely misses detecting the area where the change actually occurs. As can be seen in the third row, FC-Siam-diff, CDNet, LUNet, and IFNet all have false detections due to the shadow created by the light, with LUNet being

the most obvious example. The fourth row contains relatively more irregular building variations, and all methods perform well, but our method extracts delicate edges. The fifth row illustrates that IFNet cannot recognize large building changes effectively, while other approaches also have a large number of error detections. However, our method is capable of identifying the actual change area almost perfectly.

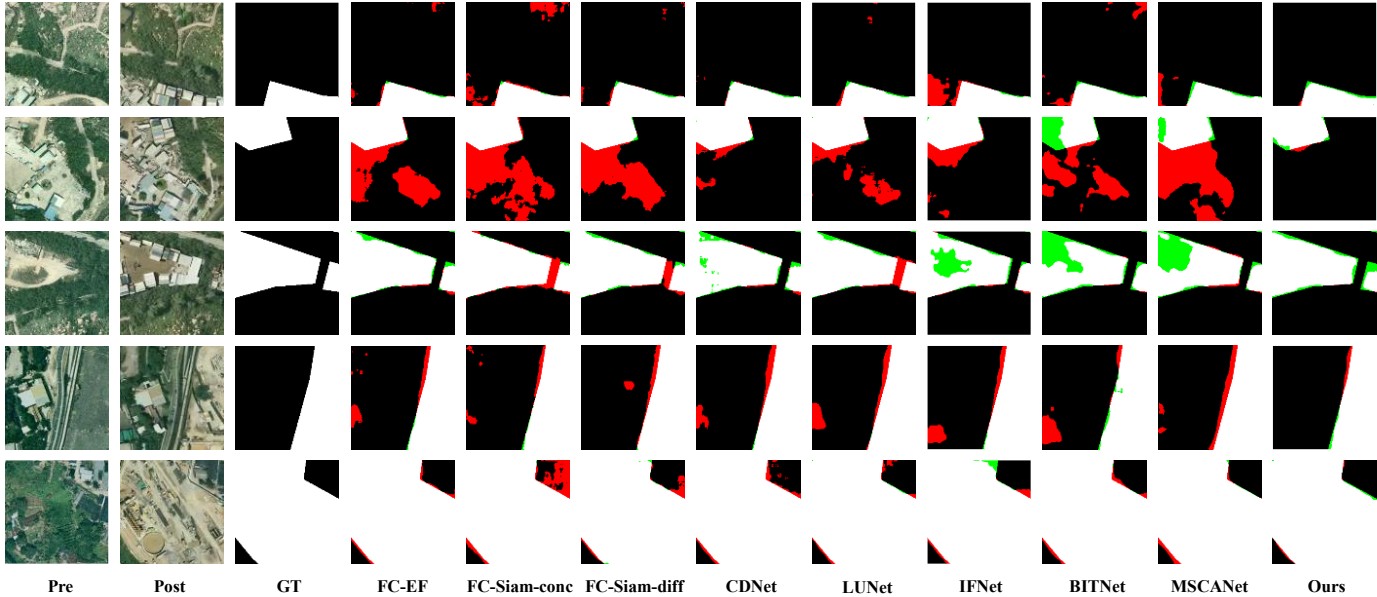

**Figure 8.** Visualization results of different CD methods in five scenarios from the SYSU-CD dataset. We plot the FP in red and the FN in green in order to better compare the CD results.

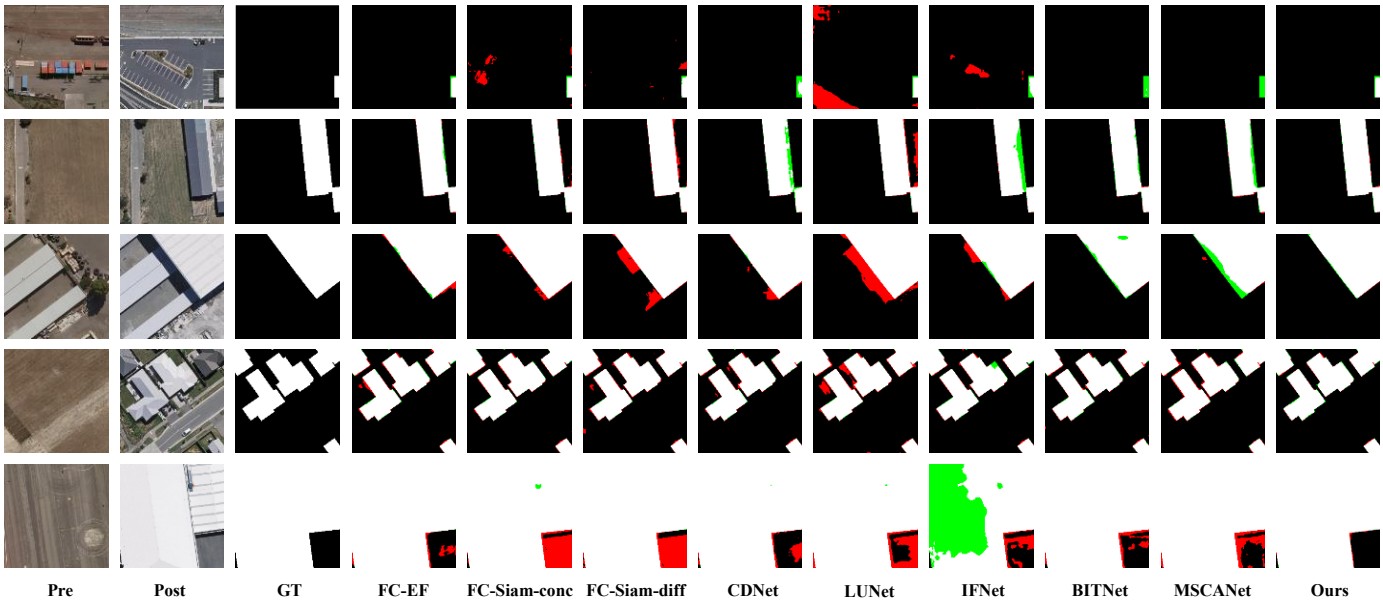

**Figure 9.** Visualization results of different CD methods in five scenarios from the WHU-CD dataset. We plot the FP in red and the FN in green in order to better compare the CD results.

Figure 10 illustrates the BCD results of various methods for five different scenarios using the S2Looking-CD dataset. Due to cloud, lighting, or shooting angle influences on this dataset, all methods do not perform as well as the first three datasets. Although there are only small building variations in the first row, the strong cloud interference leads to unsatisfactory detections by the comparison methods. The second and third rows represent

expansion scenes of buildings, which are missed by many methods (LUNet, IFNet) due to the fact that the buildings in the pre-temporal images are already under construction. Our method can detect whether buildings are newly constructed additions more effectively because our self-attention mechanism maintains the global validity of feature information. In the fourth and fifth rows, we can observe that the illumination of the post-temporal images is extremely weak, which undoubtedly adds to the difficulty of BCD. Especially in the fifth row, the changed areas are numerous and dense. There is a large number of false positive and false negative results with the other eight comparative methods. In contrast, our method distinguishes unchanged and changed areas effectively regardless of lighting conditions.

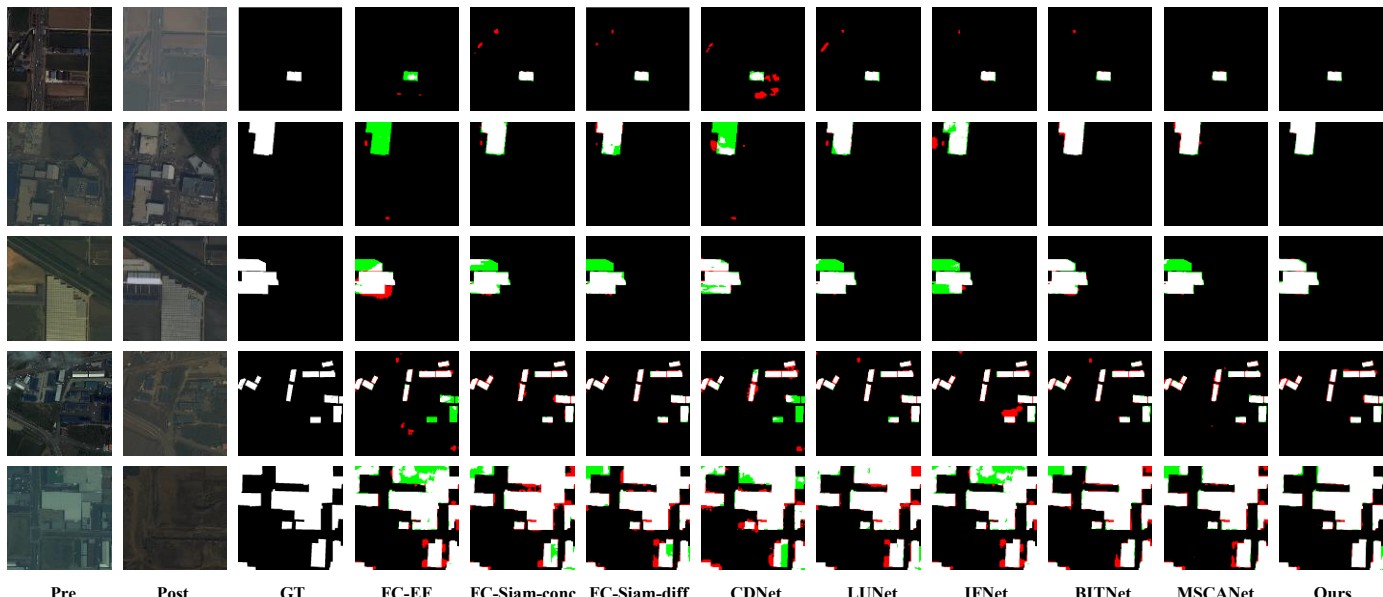

**Figure 10.** Visualization results of different CD methods in five scenarios from the S2Looking-CD dataset. We plot the FP in red and the FN in green in order to better compare the CD results.

### 3.5. Quantitative Comparative Experiments

Table 3 reports the overall comparison results of eight evaluation metrics on the LEVIR-CD dataset. Our method achieves the most inspiring results in terms of Precision (93.41%), F1-score (92.02%), IOU_0 (98.07%), IOU_1 (85.22%), mIOU (91.64%), OA (98.26%), and Kappa (91.04%). Notably, MSCANet obtains the highest Recall (91.85%), indicating that the effective combination of CNN and Transformer can improve the network's ability to perceive building changes. Our proposed method has a slightly lower Recall value than LUNet, IFNet, BITNet, and MSCANet, but our F1-score value is still 2.7% higher than BITNet, demonstrating that our method has the strongest comprehensive performance.

**Table 3.** Quantitative comparisons of various CD methods from the LEVIR-CD dataset.

| Method | Precision | Recall | F1-Score | IOU_0 | IOU_1 | mIOU | OA | Kappa |
|---|---|---|---|---|---|---|---|---|
| FC-EF | 79.91 | 82.84 | 81.35 | 95.38 | 68.56 | 81.97 | 95.80 | 78.99 |
| FC-Siam-conc | 81.84 | 83.55 | 82.68 | 95.74 | 70.48 | 83.11 | 96.13 | 80.51 |
| FC-Siam-diff | 78.60 | 89.30 | 83.61 | 95.71 | 71.84 | 83.77 | 96.13 | 81.43 |
| CDNet | 84.21 | 87.10 | 85.63 | 96.43 | 74.87 | 85.65 | 96.77 | 83.81 |
| LUNet | 85.69 | 90.99 | 88.73 | 97.13 | 79.75 | 88.44 | 97.42 | 87.28 |
| IFNet | 85.37 | 90.24 | 87.74 | 96.91 | 78.16 | 87.53 | 97.21 | 86.17 |
| BITNet | 87.32 | 91.41 | 89.32 | 97.31 | 80.70 | 89.00 | 97.59 | 87.96 |
| MSCANet | 83.75 | **91.85** | 87.61 | 96.81 | 77.95 | 87.38 | 97.13 | 85.99 |
| **Ours** | **93.41** | 90.67 | **92.02** | **98.07** | **85.22** | **91.64** | **98.26** | **91.04** |

Note that the values in bold are the highest.

Table 4 presents the quantitative comparison results of our method and the comparison methods on the SYSU-CD dataset. Among the comparison methods, FC-EF achieves the best Recall value, with a Precision and F1-score of 64.58% and 75.13%, respectively, which are 20.79% and 5.53% lower than our Precision and F1-score. Benefiting from its multi-scale feature fusion strategy, IFNet obtains an F1-score of 80.98%, exceeding ours by 0.32%. The MMDF module can reduce the loss of key information in the process of multi-scale feature fusion. Therefore, we have the highest Precision value of 85.37%, which is 4.75% higher than the second ranked BITNet.

**Table 4.** Quantitative comparisons of various CD methods from the SYSU-CD dataset.

| Method | Precision | Recall | F1-Score | IOU_0 | IOU_1 | mIOU | OA | Kappa |
|---|---|---|---|---|---|---|---|---|
| FC-EF | 64.58 | **89.79** | 75.13 | 82.21 | 60.17 | 71.19 | 85.98 | 65.73 |
| FC-Siam-conc | 65.98 | 89.39 | 75.99 | 83.08 | 61.28 | 72.18 | 86.65 | 67.04 |
| FC-Siam-diff | 70.84 | 84.87 | 77.22 | 85.24 | 62.90 | 74.07 | 88.19 | 69.34 |
| CDNet | 74.61 | 84.10 | 79.08 | 86.90 | 65.39 | 76.15 | 89.50 | 72.10 |
| LUNet | 76.14 | 81.74 | 78.84 | 87.18 | 65.08 | 76.13 | 89.65 | 72.01 |
| IFNet | 75.29 | 87.60 | **80.98** | 87.77 | **68.04** | 77.91 | 90.30 | 74.52 |
| BITNet | 80.61 | 79.29 | 79.95 | 88.46 | 66.59 | 77.53 | 90.62 | 73.83 |
| MSCANet | 79.83 | 78.50 | 79.16 | 88.04 | 65.51 | 76.77 | 90.25 | 72.80 |
| **Ours** | **85.37** | 76.45 | 80.66 | **89.45** | 67.59 | **78.52** | **91.36** | **75.12** |

Note that the values in bold are the highest.

Table 5 shows that we outperform the other eight methods in terms of Precision, F1-score, IOU_0, IOU_1, mIOU, OA, and Kappa, achieving 91.44%, 89.22%, 98.97%, 80.55%, 89.76%, 99.01%, and 88.71%, respectively, from the WHU-CD dataset. Furthermore, these results confirm the validity of the progressive context-aware aggregation and MMDR modules. Since the LSTM module in LUNet is not sensitive to building changes before and after the earthquake, its overall index is unsatisfactory. The results indicate that FC-Siam-diff achieves the highest Recall value of 94.30%, which is 7.18% higher than our method. In spite of the fact that our method does not achieve the highest Recall due to our focus on preventing false positives, our IOU_0 and IOU_1 values reach 98.97% and 80.55%, respectively, indicating that our proposed method is capable of correctly identifying unchanged and changed areas.

**Table 5.** Quantitative comparisons of various CD methods from the WHU-CD dataset.

| Method | Precision | Recall | F1-Score | IOU_0 | IOU_1 | mIOU | OA | Kappa |
|---|---|---|---|---|---|---|---|---|
| FC-EF | 70.43 | 92.31 | 79.90 | 97.72 | 66.53 | 82.12 | 97.82 | 78.77 |
| FC-Siam-conc | 63.80 | 91.81 | 75.28 | 97.04 | 60.36 | 78.70 | 97.16 | 73.83 |
| FC-Siam-diff | 65.98 | **94.30** | 77.63 | 97.33 | 63.44 | 80.38 | 97.44 | 76.32 |
| CDNet | 81.75 | 88.69 | 85.08 | 98.47 | 74.03 | 86.25 | 98.54 | 84.31 |
| LUNet | 66.32 | 93.06 | 77.45 | 97.33 | 63.19 | 80.26 | 97.45 | 76.13 |
| IFNet | 86.51 | 87.69 | 87.09 | 98.72 | 77.14 | 87.93 | 98.78 | 86.45 |
| BITNet | 82.35 | 92.59 | 87.17 | 98.66 | 77.26 | 87.96 | 98.72 | 86.50 |
| MSCANet | 83.07 | 90.70 | 86.72 | 98.63 | 76.55 | 87.59 | 98.69 | 86.03 |
| **Ours** | **91.44** | 87.12 | **89.22** | **98.97** | **80.55** | **89.76** | **99.01** | **88.71** |

Note that the values in bold are the highest.

Table 6 shows the comprehensive comparison results of all methods from the S2Looking-CD dataset. All methods achieve lower performance than those from the above three datasets due to the presence of a large number of side-looking images and irrelevant changes, such as seasonal and light variations. Moreover, the S2Looking dataset contains fewer instances of changing buildings than the three other datasets. The IOU_0 values of all methods are therefore over 98%, whereas the IOU_1 value is below 50%. Under cloud interference, CDNet's performance is poor in terms of precision, which indicates

that contraction and expansion blocks have difficulty detecting building changes. It should be noted that our Precision and F1-score values are still higher than the other comparison methods, achieving 69.68% and 65.36%. This shows that our proposed method is also capable of handling difficult datasets.

**Table 6.** Quantitative comparisons of various CD methods from the S2Looking-CD dataset.

| Method | Precision | Recall | F1-Score | IOU_0 | IOU_1 | mIOU | OA | Kappa |
|---|---|---|---|---|---|---|---|---|
| FC-EF | 45.02 | 50.28 | 47.51 | 98.66 | 31.15 | 64.91 | 98.67 | 46.84 |
| FC-Siam-conc | 60.60 | 59.88 | 60.24 | 99.05 | 43.10 | 71.08 | 99.05 | 59.76 |
| FC-Siam-diff | 60.00 | 59.63 | 59.81 | 99.04 | 42.67 | 70.85 | 99.04 | 59.33 |
| CDNet | 47.70 | 50.86 | 49.23 | 98.73 | 32.65 | 65.70 | 98.75 | 48.60 |
| LUNet | 59.95 | 58.59 | 59.26 | 99.03 | 42.11 | 70.57 | 99.04 | 58.78 |
| IFNet | 59.93 | 57.66 | 58.77 | 99.03 | 41.62 | 70.32 | 99.03 | 58.28 |
| BITNet | 64.96 | **62.95** | 63.94 | 99.14 | 47.00 | 73.07 | 99.15 | 63.51 |
| MSCANet | 64.63 | 57.67 | 60.95 | 99.11 | 43.84 | 71.47 | 99.12 | 60.51 |
| **Ours** | **69.68** | 61.54 | **65.36** | **99.21** | **48.54** | **73.88** | **99.22** | **64.96** |

Note that the values in bold are the highest.

Figure 11 illustrates the results of various methods for eight indicators in the form of cumulative distribution curves. LUNet, IFNet, BITNet, and our method perform exceptionally well, with all eight index values ahead of the other comparison methods. We outperform all comparison methods except the Recall metric. Although LUNet, IFNet, BITNet, and MSCANet narrowly outperform our method in the Recall metric, our Precision metric is 5% higher than even the second place BITNet. Additionally, the highest F1-score value indicates that we are capable of achieving comprehensive BCD results.

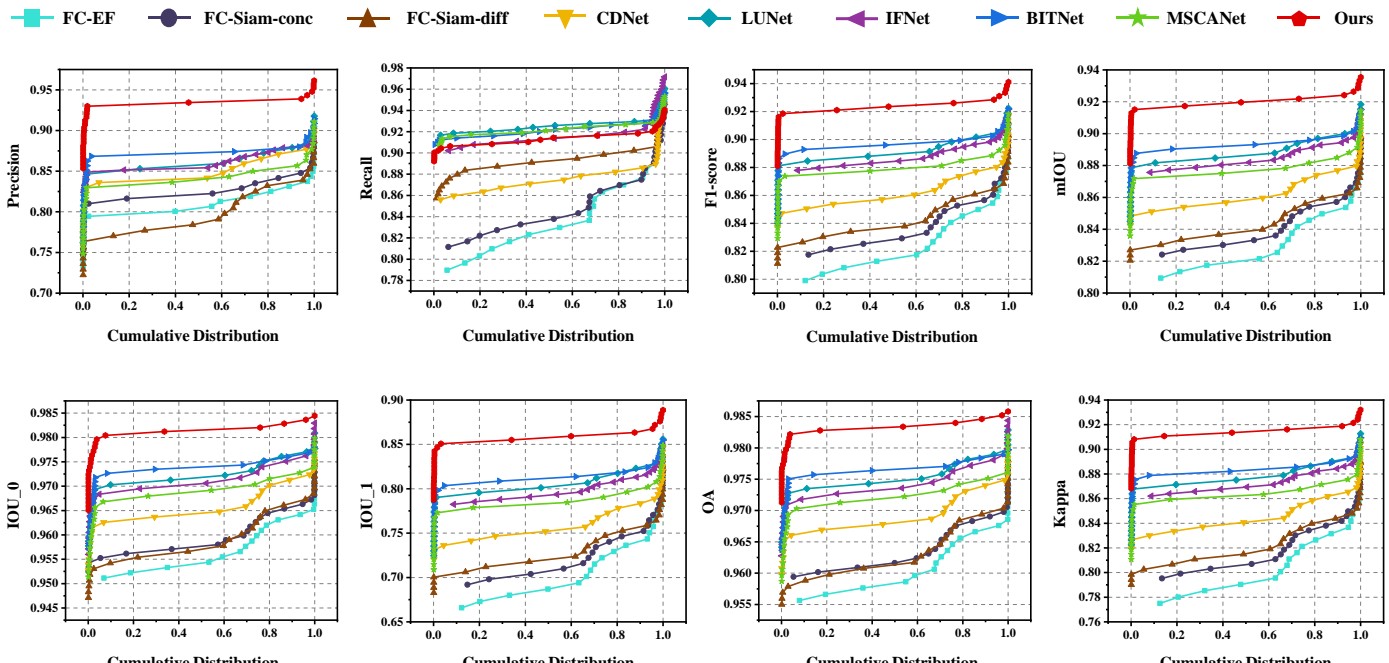

**Figure 11.** Eight metrics results of cumulative distribution curves for 921 images from the LEVIR-CD dataset.

Figure 12 illustrates the results of eight metrics in box diagrams. Each box has a horizontal line that represents the midline. The lines below and above the box indicate the minimum and maximum values, respectively. According to the box diagrams, the Precision metric distribution has the greatest difference. This suggests that our method is most effective at distinguishing actual changing pixels. Despite our lower Recall value than

LUNet, IFNet, BITNet, and MSCANet, the discrepancy is not very significant. Additionally, we find that the data distributions for the other six metrics are ideal, exceeding those of other methods by a wide margin. Therefore, our proposed method demonstrates a high degree of reliability among the nine methods tested.

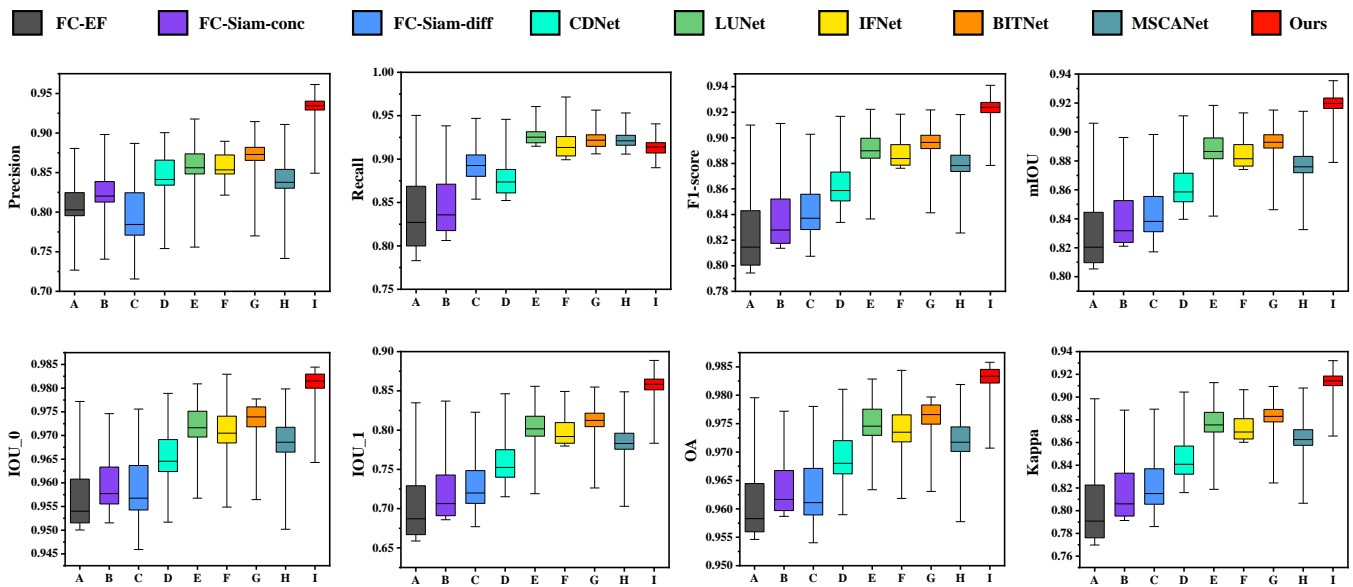

**Figure 12.** Eight metrics results of box diagrams for 921 images from the LEVIR-CD dataset. The nine alphabets A-I on the horizontal axis represent FC-EF, FC-Siam-conc, FC-Siam-diff, CDNet, LUNet, IFNet, BITNet, MSCANet, and our method, respectively.

### 3.6. Computational Efficiency Experiment

To assess the efficiency of various methods, we use two metrics: the number of parameters (Params) and floating points of operations (FLOPs) to calculate the computational efficiency of various methods. Note that as the number of Params and FLOPs of the model decreases, the complexity and computation cost of the model decrease as well.

Each method is tested on two images of the same size ($1 \times 3 \times 224 \times 224$ pixels), and comparative results are presented in Table 7. Due to the simplicity of the model, the parameters for the three FC-based methods as well as CDNet are the smallest. BITNet's light backbone also contributes to its impressive results in terms of model efficiency. Because of the deep stacking convolutional networks, IFNet performs poorly in terms of model efficiency. We note that both our Parms and FLOPs values are the highest, reaching 61.41 M and 77.01 G, respectively. Due to the deep convolutions and large global receptive fields of our method, we are compelled to increase its capacity.

**Table 7.** Analysis of the computational efficiency of various methods.

| Method | Params(M) | FLOPs(G) |
|---|---|---|
| FC-EF | 1.35 | 2.74 |
| FC-Siam-conc | 1.54 | 4.08 |
| FC-Siam-diff | 1.35 | 3.62 |
| CDNet | 1.43 | 17.97 |
| LUNet | 8.45 | 13.27 |
| IFNet | 35.73 | 62.98 |
| BITNet | 3.01 | 6.51 |
| MSCANet | 16.42 | 11.33 |
| Ours | 61.41 | 77.01 |

## 4. Discussion

This paper describes the development of a progressive context-aware aggregation module capable of extracting local–global feature information from bi-temporal images. We investigate ways to stack the two so as to utilize the strengths of both deep convolution and self-attention. We can see that in four different combinations of ablation experiments, deep convolution always comes first. This is because during the initial stage of feature extraction, the network mainly focuses on shallow local change information, where deep convolution is effective. Self-attention has the advantage of capturing the correlation between the long spatial and temporal positions of different features, which is critical for extracting high-level semantic information. Our final embedding combination is C-C-T-T, which means that we extract the superficial representation information of changing buildings through two layers of deep convolution, and then apply two layers of self-attention to obtain high-level semantic information about the changing buildings, while ensuring global validity of the feature information.

Figure 13 illustrates the attention maps generated by the four stages of the progressive context-aware aggregation module. Stage1 and Stage2 employ deep convolution to obtain shallow change features, while Stage3 and Stage4 utilize self-attention to excavate deep change semantic information. Deep convolution is capable of extracting shallow change features regardless of small or large changes. In this way, we will be able to locate and focus on the actual area of change. Integrating the a priori change information into Stage3 and Stage4 will further enhance the global validity of the feature information.

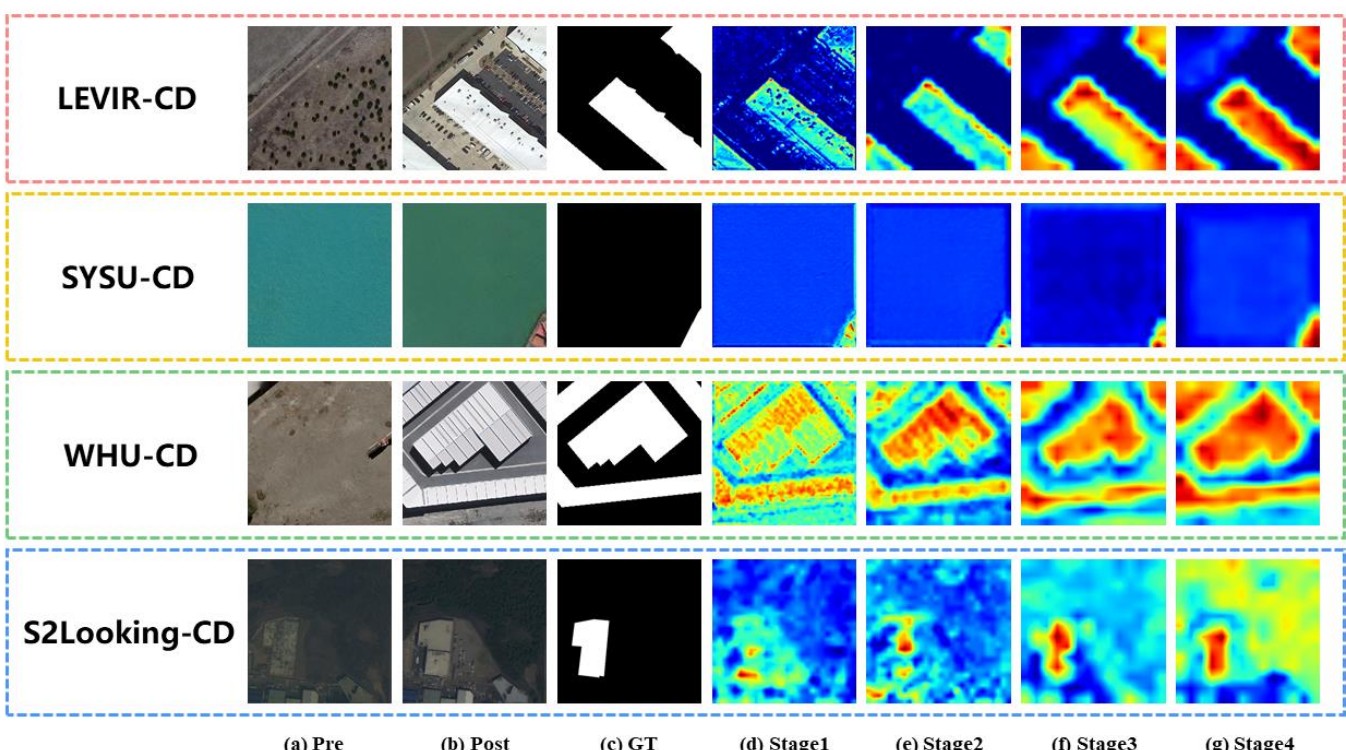

|  | (a) Pre | (b) Post | (c) GT | (d) Stage1 | (e) Stage2 | (f) Stage3 | (g) Stage4 |

**Figure 13.** Visual representations of the attention maps generated by the four stages. Note that the color blue indicates a lower level of attention, whereas the color red indicates a higher level of attention.

There is no doubt that in all examples of the four datasets, the progressive context-aware aggregation module detects actual changes in the image from shallow to deep levels. Finally, all attention is focused on the changing area. Even though these analyses serve as post hoc CD explanations, they may provide evidence that our proposed method is based on credible information and that CD predictions are based on factors relevant to buildings.

## 5. Conclusions

In this paper, we propose an effective BCD method. The progressive context-aware aggregation module enables the network to extract rich local–global feature information from bi-temporal images more effectively through the reasonable stacking of deep convolution and self-attention. In order to make the process of fusing feature information more reasonable, the MMDR module can group the extracted feature information based on pre- and post-temporal sequences, and learn the key change information of the prior groups through multi-level dense fusion. Extensive experimental results demonstrate that our proposed method outperforms other eight methods on the LEVIR-CD, SYSU-CD, WHU-CD, and S2Looking-CD datasets. In each of the four datasets discussed above, our precision values reached 93.41%, 85.37%, 91.44%, and 69.68%, respectively.

The results obtained with the proposed method have been demonstrated to be inspiring. Nevertheless, all of these results are based on labeled datasets, which are very labor- and time-intensive to collect and label. In contrast, unlabeled data are easier to obtain, so our main research focus will be on performing our method on unlabeled data with self-supervised BCD.

**Author Contributions:** Conceptualization, C.X., Z.Y. (Zhaoyi Ye), L.M. and W.Y.; methodology, Z.Y. (Zhiwei Ye); software, L.M.; validation, L.M., S.S. and Y.H.; formal analysis, W.Y.; investigation, L.M.; data curation, W.O.; writing—original draft preparation, Z.Y. (Zhaoyi Ye); writing—review and editing, C.X.; visualization, Y.H.; supervision, C.X., W.Y. and L.M.; project administration, C.X.; funding acquisition, C.X. and W.Y. All authors have read and agreed to the published version of the manuscript.

**Funding:** This research was funded by National Natural Science Foundation of China (Nos. 41601443); Scientific Research Foundation for Doctoral Program of Hubei University of Technology (BSQD2020056); Science and Technology Research Project of Education Department of Hubei Province (B2021351); Natural Science Foundation of Hubei Province (2022CFB501); University Student Innovation and Entrepreneurship Training Program Project (202210500028).

**Data Availability Statement:** Not applicable.

**Conflicts of Interest:** The authors declare no conflict of interest.

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
