# Peer review of "Progressive Context-Aware Aggregation Network Combining Multi-Scale and Multi-Level Dense Reconstruction for Building Change Detection"

_remotesensing, doi:10.3390/rs15081958_

Round 1

Reviewer 1 Report

This paper proposes a network model for building detection, and tests the model using four publicly available datasets, and the accuracy of the architecture is better than that of other models. The content of the paper is informative, the experiment is sufficient, and the description is clear. I have three  questions to discuss with the author.

1. The data of S2Look-CD is divided into 512x512 during training, but other data is divided into 256x256, what is the reason?

2. The resolution of the four datasets is close, about 0.5 meters, if you use other resolution data to test, can you still maintain a high generalization ability?

3. It is recommended to add the training loss function change curve in the paper to prove that the model has not been overfitted.

Reviewer 2 Report

Introduction

This explains why this is a useful and apposite topic to research. It gives a brief overview of the topic and a slightly deeper review of current DL methods.

In the statement of contributions, can they claim no. 3 as a contribution, since it’s more a verification of nos. 1 and 2?

Materials and methods

The section gives an overview of the architecture then a deeper description of the two novel components. There are some minor queries.

Fig 1: the connection betweem PCAA and MMDR is inferred. Is it the two self-attention blocks feed into the pre and post blocks? We also assume the two red arrows imply definition/further enlargement?

Why are the input patches downsampled – couldn’t the input just be smaller?

It seems odd that you use high res images then downsample

Eq 9 – is this a cross entropy loss? So is (1 – Gi) correct?

Experiments and results

This section describes the data sets that have been used and the evaluation metrics. It reports the results, an ablation study and a comparison against eight other recent methods. There are some queries.

Dataset 4

Each image is 1024^2, so it can be broken into 4x 512^2 images

5,000 input images will give 20,000 512^2 pixel images?

How does this equate to 31,500 training + 4,500 valiadation + 9,000 testing = 45,000 images?

Figs 3 – 6, the captions should explain what each line of images is

How was the ground truth determined?

Line 284, can you also define OA and PE

Ablation: is it reasonable to try C-T-T-T?

Section 3.4

You have thousands of image pairs, but show the results of processing just a few. How were these 20 chosen? Are they representative? Why not present evasluation statistics for the whole set of images? (as in the following section). The result images, while impressive, do not convince us that the method is successful. I wold question whether this section is really needed.

Fig 12

What are the horizontal axes?

Line 441

Repeated word, should it be number of parameters?

The current architecture outperforms other networks on all metrics except Recall. The cost of this is the network is almost 2x the number of parameters as the next biggest, meaning ¼ more operations in processing as image

References

Is 5 a correct reference for DICE?

Reviewer 3 Report

The authors proposes a progressive context-aware aggregation network combining multi-scale and multi-level dense reconstruction to identify detailed texture-rich building change information. The paper is interesting, and it clearly presented the methods and results, but the discussion of the results needs an improvement, and needs to be compared with the literature with more criteria. Also, the language must be improved.

Round 2

Reviewer 3 Report

The paper is improved, and it can be accepted.